# Tumor-to-Tumor Metastases Involving Clear Cell Renal Cell Carcinomas: A Diagnostic Challenge for Pathologists Needing Clinical Correlation †

Claudia Manini [1,2], Claudia Provenza [1], Leire Andrés [3], Igone Imaz [3], Rosa Guarch [4], Raffaelle Nunziata [1] and José I. López [5,*]

1   Department of Pathology, San Giovanni Bosco Hospital, 10154 Turin, Italy
2   Department of Sciences of Public Health and Pediatrics, University of Turin, 10124 Turin, Italy
3   Department of Pathology, Cruces University Hospital, 48903 Barakaldo, Spain
4   Department of Pathology, Hospital Virgen del Camino, 31008 Pamplona, Spain
5   Biomarkers in Cancer Unit, Biocruces-Bizkaia Health Research Institute, 48903 Barakaldo, Spain
*   Correspondence: joseignacio.lopez@biocrucesbizkaia.org
†   This paper is dedicated to Prof. Francisco Colina-Ruizdelgado, a pathologist and a teacher of pathologists at the 12 de Octubre Hospital in Madrid, Spain, for his tireless advices to focus on the clinical-pathological correlation.

**Abstract:** Tumor-to-tumor metastasis is a rare event which it is specifically up to pathologists to bring to light correctly. The histological identification of such tumor-to-tumor cases is simple when the respective histologies are different but can be problematic if the case includes two carcinomas with similar cytoarchitecture viewed one inside the other under the microscope. We report four cases of this condition in which clear cell renal cell carcinoma is involved, either as a receptor or as a donor, and remark on the difficulties in recognizing some of them. Appropriate clinical–pathological correlation, including a review of the patient's antecedents and radiological exams, would be a great help in routinely identifying tumor-to-tumor metastases.

**Keywords:** tumor-to-tumor metastasis; clear cell renal cell carcinoma; tumor sampling; clinical-pathologic correlation; histopathology; immunohistochemistry





## 1. Introduction

Tumor-to-tumor metastasis is a rare event in clinical practice that sometimes involves clear cell renal cell carcinomas (CCRCC), either as donors [1–6] or receptors [7–9]. Often, the two components are identified making the case somewhat exotic. However, this may lead to conflicting diagnostic situations and thus to incomplete pathological reports when the donor and the receptor are carcinomas with compatible histologies, because their morphological variability under the microscope may be misunderstood as a sort of intratumor heterogeneity (ITH). Correct identification in these particular cases calls for not only personal experience and attitude, but also the knowledge of the previous history of the patient, free access to radiological exams/reports, and, finally, appropriate clinical–pathological correlation.

This article brings together four clinical–pathological cases of tumor-to-tumor metastasis in which CCRCC is directly involved, twice as a donor and twice as a receptor. The authors seek to highlight the responsibility of pathologists, the only doctors in the clinical practice who are able to detect tumor-to-tumor metastasis, for clinically correlated histological analysis of tumors in general and CCRCC in particular. The growing number of tumor-to-tumor metastases reported in the last 5 years confirms that this situation is more frequent than previously thought [10–20].

## 2. Case Reports

**Case 1.** *CCRCC metastasized by breast carcinoma.*

A renal mass was discovered in a 67-year-old woman during a follow-up on a breast duct carcinoma diagnosed three years before. On the CT scan, the tumor was found to be located in the lower pole of the left kidney. It measured 4.5 cm in maximum diameter (Figure 1A, white arrow).

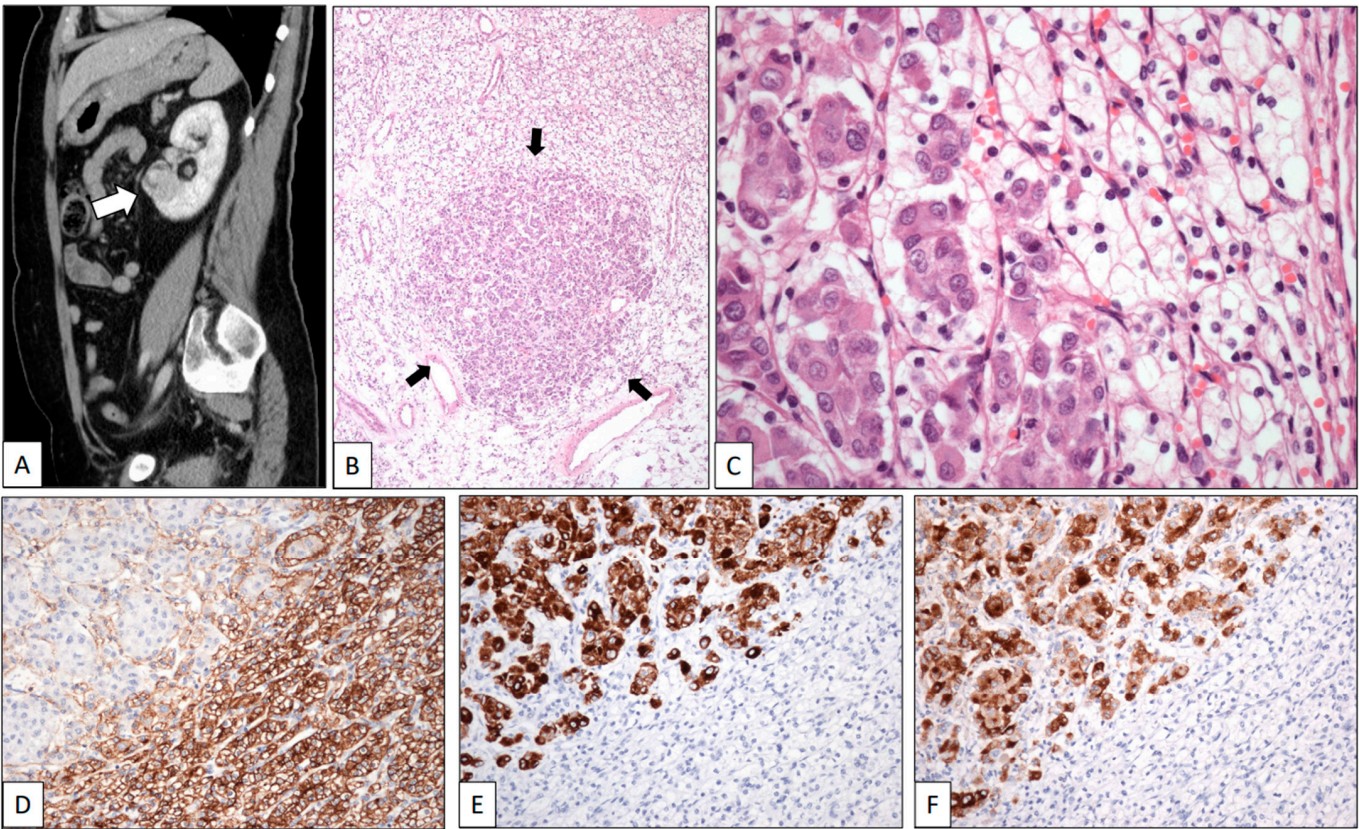

**Figure 1.** Clear cell renal cell carcinoma metastasized by breast carcinoma. (**A**) Sagittal CT scan shows a tumor in the lower pole of the left kidney (white arrow). Low-power view of one of the multiple metastases of the surgical specimen showing a nest of eosinophilic cells (black arrows) in a background of clear cells (**B**). High-power detail of a micrometastasis (arrows) showing large eosinophilic cells of breast carcinoma intermingled with small clear cells of renal cell carcinoma (**C**). The immunohistochemical profile of the two components displays positive staining with CD10 (**D**) in the clear cell renal cell carcinoma whereas breast cancer cells were positive with CK19 (**E**) and GCDFP-15 (**F**).

A left radical nephrectomy was performed. Histologically, a pT2 Grade 1 typical CCRCC was observed. In the low-power view, the neoplasm showed two different cell types (Figure 1B). Small-sized tumor cells arranged in sheets and nests showing clear cytoplasm and small dark nuclei without nucleoli were admixed with small nests of large cells displaying eosinophilic cytoplasm and large nuclei with conspicuous nucleoli (Figure 1C). CD10 (Figure 1D) and PAX-8 were positive for CCRCC, whereas progesterone and estrogen receptors, CK7 (Figure 1E), and GCDFP-15 (Figure 1F) stained the breast carcinoma. The patient was alive at the last contact, two years after the diagnosis of the kidney tumor.

**Case 2.** *CCRCC metastasizing into a meningioma.*

A 79-year-old man was referred due to memory disturbances. Clinical antecedents included a pT2, Grade 2 CCRCC treated with right total nephrectomy 8 years before.

The patient's follow-up was unremarkable. MR showed a solid extra-axial fronto-basal lesion with irregular margins and a posterior hypo-intense signal (Figure 2A, white arrow). At the time of neurosurgery, a small peripheral fragment of the mass was submitted to intra-operative analysis, and a diagnosis consistent with meningioma was given. The gross specimen submitted was a 5 cm yellow-brownish nodule 5 cm in diameter (red arrow) with peripheral whitish areas (blue arrow) (Figure 2B). The histological study revealed a neoplasm with two components: a low-grade clear cell carcinoma, on one hand, and a whorled tumor with nuclear pseudoinclusions, on the other (Figure 2C). With immunohistochemistry, PAX-8, carbonic anhydrase IX (Figure 2D), and CD10 (Figure 2E) were positive in the clear cell carcinoma, revealing its renal origin. By contrast, EMA (Figure 2F) and progesterone receptor (Figure 2G) were positive in the whorled component confirming its meningothelial origin. The patient is alive and without residual disease after 6 months of follow-up.

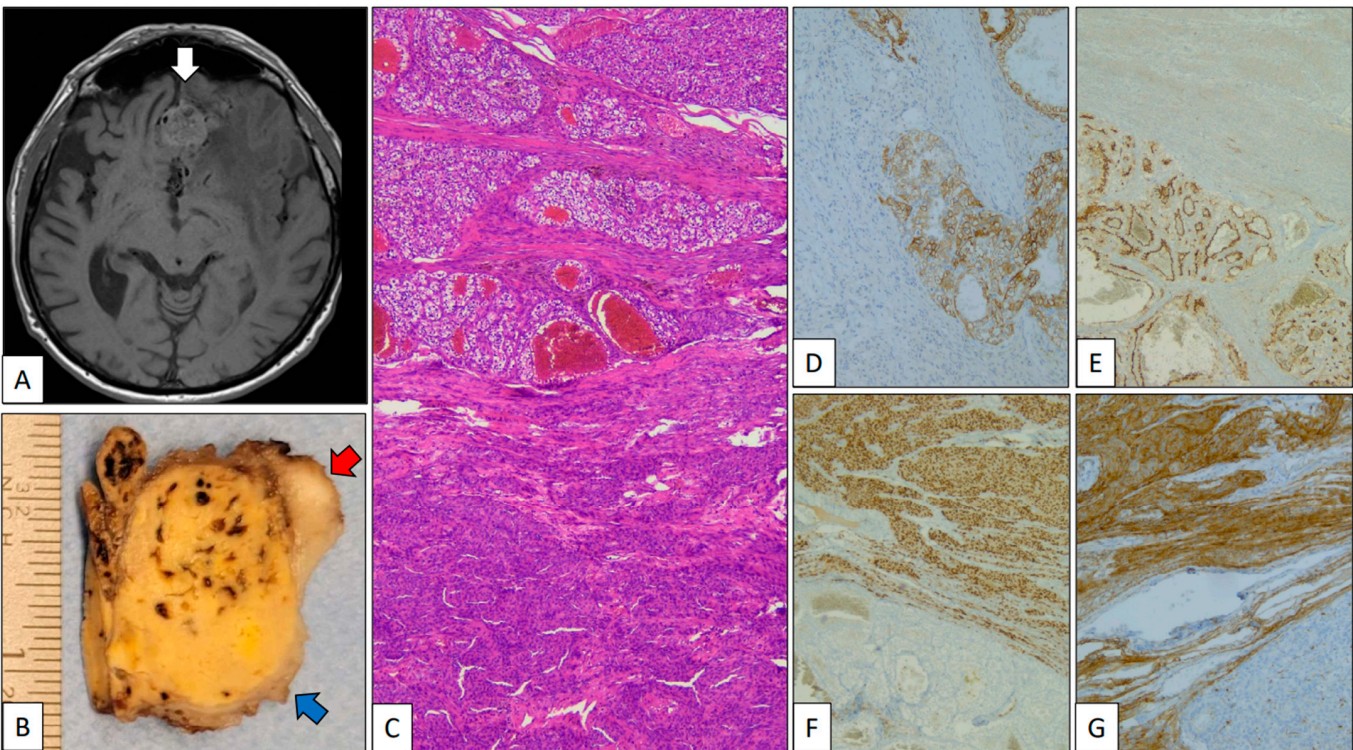

**Figure 2.** Clear cell renal cell carcinoma metastasizing in meningioma. (**A**) Axial MR of the brain showing an extra-axial heterogeneous fronto-basal tumor mass (white arrow). (**B**) Partial view of a part of the gross specimen showing a mixed yellow-brownish (clear cell renal cell carcinoma, blue arrow) and whitish (meningioma, red arrow) appearance. (**C**) Panoramic view of the interface between clear cell renal cell carcinoma (top) and meningioma (bottom). Positive immunostaining with carbonic anhydrase IX (**D**) and CD10 (**E**) in the clear cell renal cell carcinoma and progesterone receptor (**F**) and EMA (**G**) in the meningioma.

**Case 3.** *CCRCC metastasizing into a papillary thyroid carcinoma, follicular variant.*

A 42-year-old man was referred due to hematuria. CT scans revealed a heterogeneous solid-cystic right renal tumor mass 12 cm in diameter and a synchronic nodule 5 cm in diameter in the right lobe of the thyroid. A right radical nephrectomy was performed and a Stage pT3a Grade 4 conventional CCRCC infiltrating the perinephric fatty tissue was histologically diagnosed. In the same hospitalization episode, a fine-needle aspiration cytology study of the thyroid nodule revealed abundant colloid material with very scarce uneventful follicular-type cells. A second CT scan after 11 months of follow-up showed that the thyroid nodule had become heterogeneous and exhibited significant growth towards the thoracic

cavity, reaching 9 cm in diameter (Figure 3A,B, white arrows). A total thyroidectomy was performed. The right dominant nodule was completely capsulated and showed two very different cellular components under the microscope (Figure 3C). The predominant part was composed of a microfollicular growth of follicular-type cells with nuclear changes diagnostic of well-differentiated papillary carcinoma (nuclear enlargement, grooving, overlapping, and optically empty nuclei) (Figure 3D). As expected, Thyroglobulin and TTF-1 (Figure 3E) were positive in this tumor component. Inside the capsule, the second component of the nodule comprised large epithelial cells with clear cytoplasm and prominent nucleoli arranged in solid lobes and accompanied by a dense mixed inflammatory infiltrate (Figure 3F). CD10, PAX-8, and carbonic anhydrase IX (Figure 3G) positivity confirmed the renal origin of this component. The patient died in the course of a disseminated disease 5 years after the initial diagnosis.

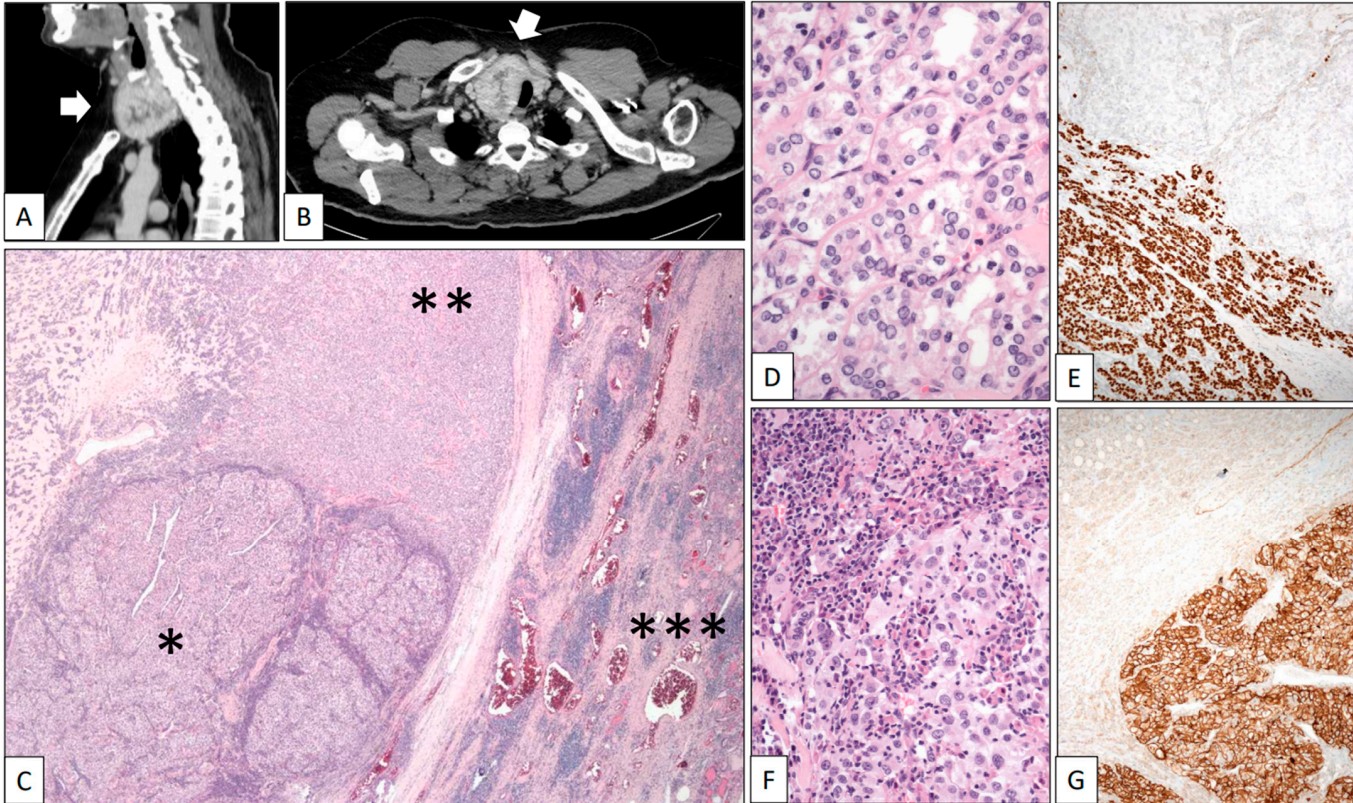

**Figure 3.** Clear cell renal cell carcinoma metastasizing in papillary carcinoma of the thyroid. (**A**) Sagittal and (**B**) axial CT scans showing a large heterogeneous tumor mass in the thyroid (white arrows). (**C**) Panoramic view of the surgical specimen displaying a tumor mass (left-lower quadrant corresponding to the renal cell carcinoma *) within a bigger mass (corresponding to the papillary carcinoma **) growing in the thyroid gland (right side ***). High-power detail showed a follicular variant of papillary thyroid carcinoma with characteristic nuclear features (**D**) positive with TTF-1 (**E**) and an inflamed renal cell carcinoma (**F**) positive with carbonic anhydrase IX (**G**).

**Case 4.** *CCRCC infiltrated by chronic lymphocytic leukemia.*

A 74-year-old woman with a previous history of B-cell chronic lymphocytic leukemia was referred due to recurrent hematuria. The initial clinical work-up sought to rule out a possible urological leukemic involvement. However, a CT scan revealed an intrarenal tumor mass 5 cm in diameter. A left total nephrectomy was then performed. The histological study showed a pT2 Grade 2 conventional CCRCC (Figure 4A). The neoplasm was composed of sheets and cords of clear cells with large clear cytoplasm and regular nuclei and occasional nucleoli within a hemorrhagic stroma (Figure 4B). A subtle infiltration of

mature lymphocytes was also found among the neoplastic epithelial cells (Figure 4B). This lymphocytic component showed a CD19, CD20, CD79α (Figure 4C), CD5 (Figure 4D), and CD23 immunophenotype, with kappa restriction, congruent with leukemic infiltration within the CCRCC. The patient is alive after 1 year of follow-up.

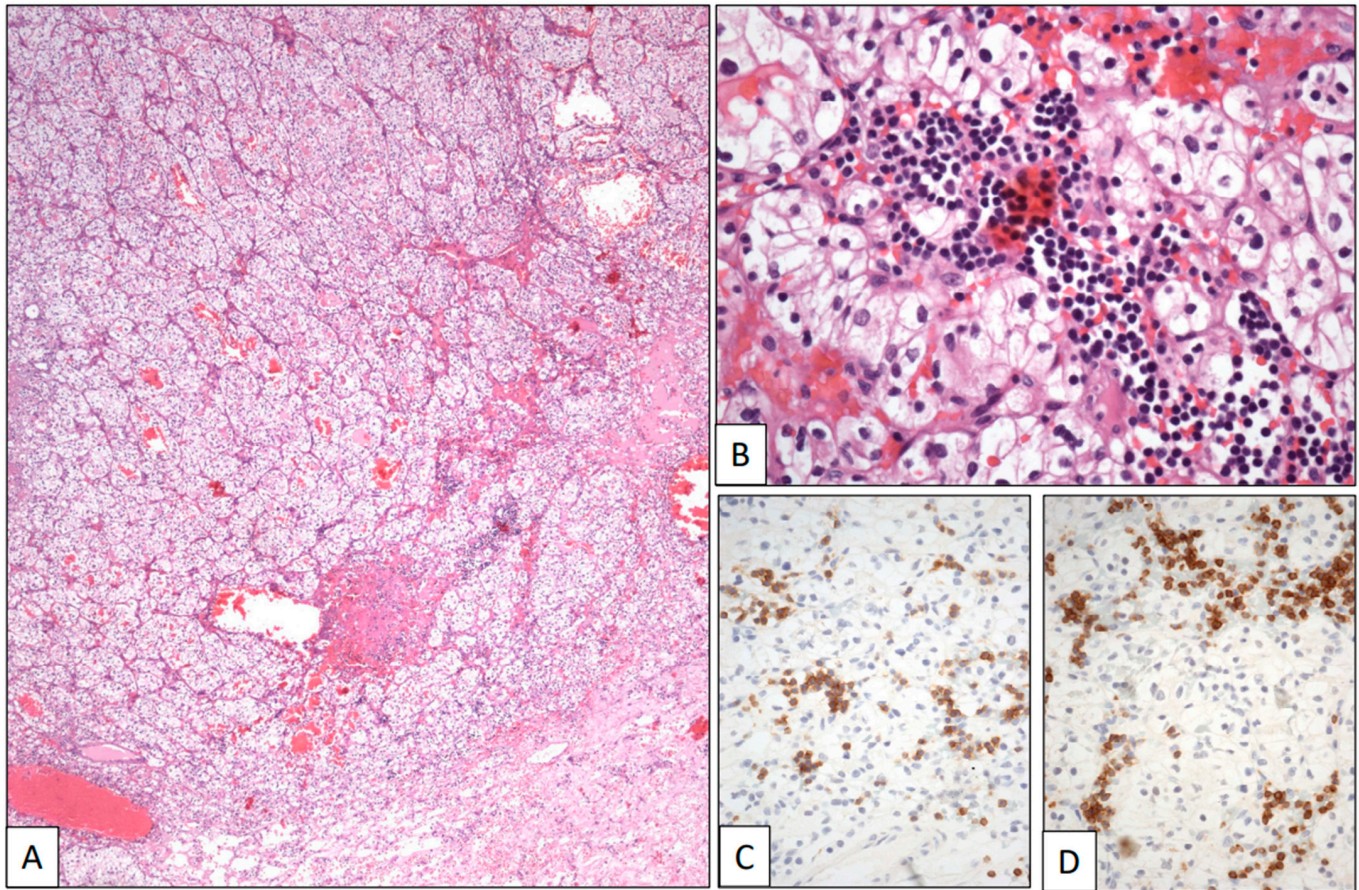

**Figure 4.** Clear cell renal cell carcinoma infiltrated by chronic lymphocytic leukemia. (**A**) Panoramic view of a conventional low-grade clear cell renal cell carcinoma. (**B**) Histological detail showing neoplastic clear cells intermingled with a small-celled monotonous lymphocytic infiltration showing positive immunostaining with CD5 (**C**) and CD79α (**D**).

## 3. Discussion

Metastatic dissemination is a major characteristic of aggressive behavior in malignant tumors and the main cause of death in cancer patients. In the era of personalized medicine, its correct identification is a must and a measure of good medical practice. Tumor-to-tumor metastasis, although rare in quantitative terms, may represent a qualitative oncological problem in whose resolution pathologists must be key doctors.

At least theoretically, any malignant tumor can metastasize into any site, tumors included. Some tumor types are more prone than others to tumor-to-tumor metastasis, which gives rise to a varied spectrum of difficulties for pathologists. When tumor-to-tumor metastasis is the first manifestation of the primary tumor, the pathologist is trapped in a labyrinth if there are no clinical antecedents. From a strict histological perspective, the identification of a papillary thyroid carcinoma metastasizing into a uterine leiomyoma [21] or a benign nerve sheath tumor [22], or a breast carcinoma into a Warthin tumor [23], or a gastric adenocarcinoma into a lipoma [24], for example, is an easy—although surprising— visual exercise because the respective histologies of donors and receptors are very different. Something similar happens under the microscope when an epithelial neoplasm metastasizes into a meningioma, as in case #2 here. This clinical situation is much more frequent than

previously thought [25,26] probably related to the high vascularization that characterizes meningiomas. Carcinomas from the digestive tract [27,28], kidney [2,3], breast [29–32], and lung [31–33], among others, have so far been reported as metastasizing into meningiomas.

CCRCC is a special example in this tumor-to-tumor setting because it is equally frequent as a donor and as a receptor. No doubt, the prominent vascularization generated in this tumor by the *VHL* gene malfunction facilitates both tumor cell arrivals and departures. This article is based on four cases of CCRCC in very different clinical contexts.

The first case—a breast carcinoma into a CCRCC—may represent a diagnostic problem under the microscope because CCRCC is a paradigmatic example of ITH [34], and pathologists, who are used to its many faces, may overlook the tumor-to-tumor metastatic seed. Here, a simple review of the clinical history serves to put the pathologist on the correct road, with the diagnosis ultimately being confirmed with a basic immunohistochemical panel. The second case—a CCRCC into a meningioma—is easier to recognize since the two tumors are histologically quite different and meningioma is a common destination of metastasis, as previously commented [25,26]. The third case—a CCRCC into a papillary thyroid carcinoma—may cause diagnostic difficulties. Here, the renal tumor component is a high-grade neoplasm and the expected clear cell morphology is lacking. Such a case could even be considered as a tumor-to-tumor metastasis on hematoxylin-eosin sections, but the site of origin may remain unclear. Again, checking the clinical history of the patient is the best technique, followed by the correct selection of immunostainings to confirm the diagnosis. The fourth case illustrates how chronic lymphocytic leukemia involving a CCRCC may mimic the inflammatory infiltration found quite frequently in this tumor [35]. The clinical history with leukemic antecedents, the morphological monotony of the infiltrating lymphoid cells, and a basic immunohistochemical study can bring to light the nature of these lymphoid cells. Other isolated cases of tumor-to-tumor metastasis implicating lymphoid neoplasms have also been reported [36].

Case 2 in this series illustrates an important issue with practical implications, i.e., the consequences of a partial tumor sampling during the intra-operative study. The surgeon removed a peripheral sample of the tumor for intra-operative analysis, and a diagnosis of meningioma was then made. However, since the piece of tissue submitted contained no trace of CCRCC, the intra-operative diagnosis missed important information. An incomplete pathological diagnosis may also result when the receptor is large and benign (meningioma, lipoma, etc.), the donor seed is small but highly malignant (CCRCC, breast carcinoma, etc.), and the tumor mass is not sufficiently sampled. This eventuality has critical clinical implications because tumor-to-tumor metastasis may be the first manifestation of an unknown, unexpected, disseminated disease. Interestingly, a feasible strategy for overcoming incomplete tumor samplings at no extra cost has recently been proposed: the so-called multi-site tumor sampling [37,38].

With few exceptions, the receptor in a metastatic tumor-to-tumor situation is always a cytologically benign tumor (lipomas, Warthin tumors, schwannomas, meningiomas . . . ) or, at most, a low-grade carcinoma (Grade 1 CCRCCs, papillary thyroid carcinomas, pheochromocytomas, well-differentiated pancreatic neuroendocrine tumors . . . ). One possible explanation of this fact can be found in a recent research paper in which Zhao et al. [39] demonstrate that CCRCC develops metastasizing sub-clones in the tumor interior, probably as an ecological adaptative response to local bad conditions in those areas, such as a deficit of oxygen and nutrients. High-grade CCRCC frequently shows central necrotic areas indicative of hypoxia. In this particular context, high-grade tumors may not be the easiest places for new tumor settlements. By contrast, low-grade CCRCC is typically hypervascularized [35], and there is no oxygen scarcity, so such new settlements are possible there. The same reason could be applied to explain why papillary thyroid carcinoma [1,5,40,41] and meningiomas [25,26] are also preferred tumors for a metastatic seed.

## 4. Conclusions

Tumor-to-tumor metastasis is a rare event whose correct diagnosis depends almost exclusively on pathologists' expertise and attitude, associated with the clinical and radiological context. The combination of these tools will usually provide a correct diagnosis. Immunohistochemistry may provide additional help or confirmatory evidence. It should be noted that CCRCC, papillary thyroid carcinoma, and meningioma are among the most frequent receptor tumors, and breast, lung, renal, and prostate carcinomas are the most frequent donors.

**Author Contributions:** Conceptualization, C.M. and J.I.L., case analysis; C.M., C.P., L.A., I.I., R.G. and R.N.; writing, C.M., C.P., I.I., R.G., R.N. and J.I.L., review, J.I.L., editing, C.M. All authors have read and agreed to the published version of the manuscript.

**Funding:** This research received no external funding.

**Institutional Review Board Statement:** Ethical review and approval were waived for this study due to the cases belong to routine practice and are not included in any project.

**Informed Consent Statement:** The patients were informed about the potential use for research of their surgically resected tissues and manifested their consent by signing a specific document approved by the Ethical and Scientific Committees.

**Data Availability Statement:** All data are contained within the article.

**Conflicts of Interest:** The authors declare no conflict of interest.

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
