# Peer review of "Tumor-to-Tumor Metastases Involving Clear Cell Renal Cell Carcinomas: A Diagnostic Challenge for Pathologists Needing Clinical Correlation†"

_clinpract, doi:10.3390/clinpract13010026_

Round 1

Reviewer 1 Report

The presented article is particularly interesting, however prior the publication I recommend the deep text and style revision.

Topic: the article is interesting and relevant, it highlights the issue of composite tumors (tumor-to-tumor metatstasis included) that can and often is overlooked and can pose a problem for proper therapy. I agree with authors, that the correct identification of these particular cases implies not only personal experience and attitude, but also the deep knowledge of the previous history of the patient and adequate access to radiological reports and finally multidisciplinary collaboration. From the point of view of the topic, the article is suitable for publication and may have direct educational relevance.

However, the style of the text is more popular than academic, and I recommend its revision. Also unusual phrases are used in the text ("frozen study", ...)

I recommend adding a broader review of the literature and analyses of other published cases.

The image documentation is of poor quality. The radiological images are too small, in this form they have no meaningful value. Histological images lack description (arrows, annotation...). The scale bar / magnification is missing.

Many formulations are based on uncertain facts or are not properly cited.

l.50-51 The growing number of tumor-to-tumor metastasis (cit.?)

l. 154  Some tumors are more prone than others (cit.?, which tumours?, what is the basis of this claim?)

l. 172-182 The paragraph addresses the fact that tumours that are similar are harder to distinguish and tumours that are more different are less problematic. I don't think this is such a serious knowledge that it belongs in the discussion.

l. 197 Identifying composite tumors is not an intraoperative diagnostic task.

l. 205-211 Paragraph contains many scientifically uncertain formulations,  citations are missing. The sentece "The answer should be yes, but the fundamentals by which this is so are complex and not easy to find." reminded me Fermat's Last Theorem with the sentece "I have discovered a truly marvelous proof of this, which this margin is too narrow to contain." It's nice, but I recommend revising it.

l. 216-217 high-grade tumors are not “attractive” places for new tumor settlements (cit.?)

l. 226 NGS may encounter methodological limitations in the detection of compound tumors (minor tumour DNA concentration,...). Since the article does not go into the possibilities of genetic analyses, I recommend omitting this part. In any case, the conclusions should summarise the text and not contain further discussion considerations..

Author Response

Reviewer 1

Thank you very much for you interest and time reviewing this short series.

The presented article is particularly interesting, however prior the publication I recommend the deep text and style revision.

Topic: the article is interesting and relevant, it highlights the issue of composite tumors (tumor-to-tumor metatstasis included) that can and often is overlooked and can pose a problem for proper therapy. I agree with authors, that the correct identification of these particular cases implies not only personal experience and attitude, but also the deep knowledge of the previous history of the patient and adequate access to radiological reports and finally multidisciplinary collaboration. From the point of view of the topic, the article is suitable for publication and may have direct educational relevance.

However, the style of the text is more popular than academic, and I recommend its revision. Also unusual phrases are used in the text ("frozen study", ...) The English has been fully reviewed. Frozen study has been changed by intra-operative study.

I recommend adding a broader review of the literature and analyses of other published cases. Eleven more citations of this phenomenon involving different tumors and tumor sites have been included (new references 10 to 20).

The image documentation is of poor quality. The radiological images are too small, in this form they have no meaningful value. Histological images lack description (arrows, annotation...). The scale bar / magnification is missing. Figures are bigger now and arrows and asterisks mark more precisely the areas of interest.

Many formulations are based on uncertain facts or are not properly cited. The rephrased text in some paragraphs may contribute to clarification.

l.50-51 The growing number of tumor-to-tumor metastasis (cit.?) This sentence is supported by the number of references published in the last five years. Since a detailed mention to all of them is out of the scope of this manuscript, we have added 11 recent references (from 10 to 20 in the revised text) to give a more precise idea of this fact.

  1. 154  Some tumors are more prone than others (cit.?, which tumours?, what is the basis of this claim?) In this sentence we mean that some tumor types are more prone than others to develop tumor-to-tumor metastases based on the recent literature review. Since the beginning of the sentence may be confusing, we have changed it by “Some tumor types…”.
  2. 172-182 The paragraph addresses the fact that tumours that are similar are harder to distinguish and tumours that are more different are less problematic. I don't think this is such a serious knowledge that it belongs in the discussion. When facing a possible tumor-to-tumor metastatic phenomenon, the pathologist will have much more difficulties to identify it if both components are morphologically similar than if they are very dissimilar. Morphology at this point matters a lot, and we honestly consider this is a point that deserves to be mentioned as a practical problem.
  3. 197 Identifying composite tumors is not an intraoperative diagnostic task. This is absolutely true. However, a correct intraoperative diagnosis in these not so common cases may change the surgical attitude. For example, if a radiologically intracranial extra-axial mass points to a diagnosis of meningioma and the intraoperative diagnosis is an adenocarcinoma the attitude may change, and this fact depends sometimes on sampling. We wanted only to stress that.
  4. 205-211 Paragraph contains many scientifically uncertain formulations,  citations are missing. The sentece "The answer should be yes, but the fundamentals by which this is so are complex and not easy to find." reminded me Fermat's Last Theorem with the sentece "I have discovered a truly marvelous proof of this, which this margin is too narrow to contain." It's nice, but I recommend revising it. Yes, you are right. The sentence has been removed.
  5. 216-217 high-grade tumors are not “attractive” places for new tumor settlements (cit.?) Changed by “may not be the easiest places for new tumor settlements”
  6. 226 NGS may encounter methodological limitations in the detection of compound tumors (minor tumour DNA concentration,...). Since the article does not go into the possibilities of genetic analyses, I recommend omitting this part. In any case, the conclusions should summarise the text and not contain further discussion considerations. We agree. This part has been omitted.

Reviewer 2 Report

General comments: This case report/series reports on 4 cases of which clear cell renal cell

carcinoma is involved, either as a “receptor or as a donor”, and remark on the difficulties in

recognizing some of them.

Specific comments:

1.           Streamlining and proofing of the entire manuscript would benefit easier flow of the paper.  For instance, in the abstract alone,  the statement “ Since the diagnosis of a tumor inside another tumor is somehow contra-intuitive, a clinical-pathologic correlation in every case, including a review of the medical antecedents and radiological exams, should be routinely preconized among pathologists” can be re-structured further. Another example is “ The fourth case -a chronic lymphocytic leukemia involving a CCRCC- is neither easy because it could rise the possibility of an example of a renal tumor with a heavy component of tumor-infiltrating lymphocytes (TILs),..”

2.           In any or all these cases, was due diligence done to establish the primary tumor and history (ie., breast cancer, thyroid and meningioma and CLL?)  For instance, little further discussion regarding the breast cancer other than stating it was diagnosed 3 years ago, is it possible there is a new primary? Any other sites of disease? These should be clearly stated rather than assumed.

3.           Further extrapolate on this concept of “receptor” and “donor” in terms of metastases, that perhaps spans beyond identification via pathological morphological findings but rather genomic identification to better define this concept of “tumor-to-tumor” metastases.  Is this inherent in clear cell RCC or other subtypes of RCC? Or other cancers in general understanding that the unified overlap in diagnosis is in the clear cell RCC

Author Response

Reviewer 2

Thank you very much for you interest and time reviewing this short series.

General comments: This case report/series reports on 4 cases of which clear cell renal cell

carcinoma is involved, either as a “receptor or as a donor”, and remark on the difficulties in

recognizing some of them.

Specific comments:

Streamlining and proofing of the entire manuscript would benefit easier flow of the paper.  For instance, in the abstract alone,  the statement “ Since the diagnosis of a tumor inside another tumor is somehow contra-intuitive, a clinical-pathologic correlation in every case, including a review of the medical antecedents and radiological exams, should be routinely preconized among pathologists” can be re-structured further. Another example is “ The fourth case -a chronic lymphocytic leukemia involving a CCRCC- is neither easy because it could rise the possibility of an example of a renal tumor with a heavy component of tumor-infiltrating lymphocytes (TILs),..”Yes, we agree that the writing is not good enough to explain both situations. Previously to the English editing we would change both sentences as follows: The first one can be changed by “An appropriate clinical-pathologic correlation, including a review of the patient’s antecedents and radiological exams, would be of much help in the routine identification of a tumor-to-tumor metastasis”. The second one can be changed by “The fourth case illustrates how a chronic lymphocytic leukemia involving a CCRCC may mimic the inflammatory infiltration occurring quite frequently in this tumor”.

  1. In any or all these cases, was due diligence done to establish the primary tumor and history (ie., breast cancer, thyroid and meningioma and CLL?)  For instance, little further discussion regarding the breast cancer other than stating it was diagnosed 3 years ago, is it possible there is a new primary? Any other sites of disease? These should be clearly stated rather than assumed. We have not focused on the specific characteristics of the primary tumors because they were well established, treated, and followed in our hospitals, and such clinical, radiological, and pathological details would have made the narrative too wordy not impacting the central idea of our paper, that is, pathologists must be aware about the possibility of a tumor-to-tumor metastasis.
  2. Further extrapolate on this concept of “receptor” and “donor” in terms of metastases, that perhaps spans beyond identification via pathological morphological findings but rather genomic identification to better define this concept of “tumor-to-tumor” metastases.  Is this inherent in clear cell RCC or other subtypes of RCC? Or other cancers in general understanding that the unified overlap in diagnosis is in the clear cell RCC. Yes, we agree in that the concept of tumor-to-tumor metastasis spans beyond the morphology. Again, morphology and immunohistochemistry are the routine tools to rise this diagnostic possibility. Such possibility should be corroborated with the clinical and radiological antecedents (the main message of the manuscript). However, it is true that genomic identification should be performed in selected cases to ascertain the diagnosis. This is especially true in CCRCC and some other tumor entities but, in general, histology and immunohistochemistry will resolve the dilemma. Anyway, the goal of this paper is to cover the first step of suspicion.

Round 2

Reviewer 2 Report

Further formatting may be helpful though edits have vastly improved the flow and fluidity of the paper.